# Measuring Upper Limb Kinematics of Forehand and Backhand Topspin Drives with IMU Sensors in Wheelchair and Able-Bodied Table Tennis Players

**DOI:** 10.3390/s21248303

**Published:** 2021-12-12

**Authors:** Jia-Wen Yam, Jing-Wen Pan, Pui-Wah Kong

**Affiliations:** 1Physical Education and Sports Science Academic Group, National Institute of Education, Nanyang Technological University, 1 Nanyang Walk, Singapore 637616, Singapore; S150051@e.ntu.edu.sg (J.-W.Y.); NIE173748@e.ntu.edu.sg (J.-W.P.); 2Office of Graduate Studies and Professional Learning, National Institute of Education, Nanyang Technological University, 1 Nanyang Walk, Singapore 637616, Singapore

**Keywords:** biomechanics, para, impairment, disability, shoulder, elbow, wrist

## Abstract

To better understand the biomechanics of para-table tennis players, this study compared the shoulder, elbow, and wrist joint kinematics among able-bodied (AB) and wheelchair players in different classifications. Nineteen participants (AB, *n* = 9; classification 1 (C1), *n* = 3; C2, *n* = 3; C3, *n* = 4) executed 10 forehand and backhand topspin drives. Shoulder abduction/adduction, elbow flexion/extension, wrist extension/flexion, respective range of motion (ROM), and joint patterns were obtained using inertial measurement unit (IMU) sensors. The results showed clear differences in upper limb kinematics between the able-bodied and wheelchair players, especially in the elbow and wrist. For the para-players, noticeable variations in techniques were also observed among the different disability classes. In conclusion, wheelchair players likely adopted distinct movement strategies compared to AB to compensate for their physical impairments and functional limitations. Hence, traditional table tennis programs targeting skills and techniques for able-bodied players are unsuitable for para-players. Future work can investigate how best to customize training programs and to optimize movement strategies for para-players with varied types and degrees of impairment.

## 1. Introduction

Para-table tennis is a popular sport in the Summer Paralympic Games, consisting of players with both physical and intellectual impairments. Para-athletes can compete in 11 classifications (Classes) [1] where Classes 1 to 5 are for players competing in a wheelchair, and Classes 6 to 10 are for standing players. Class 11 is for players with intellectual impairment. In high-performance table tennis, coaches, and practitioners recognized comprehensive and perfect technique as a prerequisite for high-level success [2,3]. Generally, the technique determines tactical potential and the likelihood of achieving champion status [4].

Traditionally, table tennis techniques and players’ movements are analyzed using video analysis [5], high-speed cameras for slow-motion [6], and 3D motion capture systems [7]. Recently, a new generation of sensors and technologies have been developed as an alternative to classical laboratory-based assessment to analyze sports techniques and movements. For example, the inertial measurement unit (IMU) sensors can capture table tennis techniques and movements like video or optical cameras with superior portability, affordability, convenience (easy-to-use), and real-time feedback [8]. In the literature, previous studies on table tennis have placed IMU sensors in a racket to detect the types of strokes [9,10], assess the ball’s speeds and spins [11], and estimate the trajectories of the racket [12]. In addition, IMU sensors were worn on players’ playing limb to identify the kinematic parameters [13], recognize stroke motions [14,15,16,17,18,19,20], and detect shots [21].

To the best of authors’ knowledge, research on para-table tennis are very limited and all previous studies have focused on non-technical skills of para-table tennis players. For example, when comparing the psychomotor performance between elite and non-elite para-table tennis players, elite players demonstrated superior cognitive flexibility, attention control, information processing, and eye-hand coordination [22]. Lim and colleagues [23] investigated the functional performance of Classes 1 and 2 table tennis players and found that Class 1 players had lower functional reach, particularly the right tipping angle (i.e., the angle between the players’ spine from a seated position to the maximal reach position on the frontal plane). The lower functional reach of Class 1 players adversely affected their performance because they were at greater risk of not being able to return a rally due to a larger area that was out of their reach. Therefore, attempts have been made to increase the functional reach, stability, and comfort for Classes 1 and 2 players by implementing design modifications to the wheelchair [24,25]. Furthermore, the mobility performance of para-table tennis players was examined by Zemková and colleagues [26] and they showed that peak and mean velocities in the acceleration phase of trunk rotation were associated with lumbar curvature and pelvic tilt angles. In other words, para-players had slower trunk rotations due to their limited trunk rotational range of motion (ROM) compared to able-bodied (AB) players. Finally, studies on para-players with intellectual impairment also focused on non-technical skills, such as cognitive profiles [27], the relationship between cognition, tactical proficiency [28], and the technical or tactical ability of players [29,30]. There are currently limited studies examining the biomechanics of the techniques between para-table tennis and AB players.

Singapore is one of the most successful countries for para-table tennis, consistently achieving podium results at international competitions. For example, Singaporean Classes 1 and 2 players clinched silver and bronze medals at the 2015 ASEAN Para Games (APG) and gold medals at the 2017 APG. Research on the technical skills of para-players will enable coaches and sport practitioners to better understand their performance because the game characteristics [31], technical, and tactical proficiencies among AB and various Classes of para-players are different. There are currently limited data available regarding the techniques between para and AB players, despite the speculation that para-players would use alternative biomechanics to compensate for their impairments [32]. Since previous studies investigating AB emphasized the importance of the upper limb to generate high racket speed [33,34], the upper limb kinematics of wheelchair table tennis players are expected to differ from those of AB. Therefore, the purpose of this study was to profile the kinematics of the playing limb in wheelchair table tennis players and to compare the findings to a group of AB while executing the forehand and backhand topspin drives. IMU sensors were used because of their portability and ease of use to facilitate data collection in the athletes’ training venues. It was hypothesized that the upper limb motion patterns of para-players would be distinct from AB while executing the selected table tennis drives.

## 2. Materials and Methods

### 2.1. Participants

This study was carried out following the Declaration of Helsinki, and the methods were approved by the Nanyang Technological University Institutional Review Board (Protocol Number: IRB-2017-08-044). All participants provided written informed consent to volunteer for the study. Parental consent was also sought for minor participants who were under 21 years old.

A total of 19 male table tennis athletes were recruited (Table 1), including 10 wheelchair players from the Table Tennis Association for the Disabled (Singapore) and 9 tertiary team players. The inclusion criteria for eligible wheelchair participants were (1) male; (2) aged between 18 and 70 years old; (3) a medically certified physical impairment; (4) classification in para-table tennis; (5) member of the Table Tennis Association for the Disabled (Singapore); (6) trained at least two sessions per week in the past three months; and (7) competed in competitions at national level or higher. The inclusion criteria for AB were: (1) male; (2) aged between 18 and 35 years old; (3) no impairment; (4) member of a polytechnic or university table tennis team; (5) trained at least two sessions per week in the past three months; and (6) competed in competitions at inter-school level or higher. For both groups, participants would be excluded if they had any surgery or severe injury to the upper body in the past six months or were experiencing discomfort or pain at the time of the study.

### 2.2. Gycroscopic 3DSuit Inertial Motion Capture System

The gyroscopic 3DSuit inertial motion capture system (Inertial Labs, Paeonian Springs, VA, USA) was adopted in the present study to capture the shoulder, elbow, and wrist joint motions of all 19 participants. This system is consisted of a Men’s Under Armor HeatGear^®^ long sleeve compression shirt and leggings (Under Armour, Inc., Baltimore, MD, USA) mounted with 17 inertial 3D orientation sensors inserted into fixed pockets over specific anatomical locations. Three sensors (i.e., sensor numbers 13 to 15) from the playing limb were interested in this study (Figure 1a). Each 3D orientation sensor contained tri-axial gyroscopes, accelerometers, and magnetometers with a mass of 0.012 kg and a dimension of 0.0564 m (length) × 0.0145 m (width) × 0.0092 m (height). According to the manufacturer’s specifications, the system had a high accuracy 1° for angle measurement. This IMU system is flexible and portable, providing researchers with compelling advantages to collect data outside the laboratory with practically unlimited capture area.

One researcher (JWY) assisted all participants to wear the 3DSuit for good consistency in fitting the IMU sensors on the athletes. After checking the position of each sensor, which should be on the appropriate anatomical landmark, a Velcro strap was tightened to secure the sensors. The IMU sensors were connected with the long sleeve compression shirt and leggings via the Sensor Bus Splitter (SB-Splitter). The SB-Splitter supported joining upper and lower body chains of sensors into a single data acquisition and transmission unit. The SB-Splitter was then connected to the Sensor Bus Universal Serial Bus Control Unit (SB-CU-USB), a data acquisition system that provided power, received, and transmitted data to a computer simultaneously via a standard mini-universal serial bus (USB) cable (Figure 1b).

Before commencing all trials, participants were required to face south in an initialization posture to calibrate the system. This prerequisite posture entailed them standing in the anatomical position with face directed forward, arms at the side with palms facing forward, and feet shoulder-width apart to reset sensors 13–15 to 0°. Subsequently, participants moved to their ready position before executing the table tennis techniques. Next, data recording was initiated and terminated using the AnimaDemo program (version 11.6) on a computer. Finally, participants’ motions were reconstructed into three-dimensional real-time movements in the AnimaDemo program at a sampling rate of 60 Hz.

### 2.3. Table Tennis Equipment

An International Table Tennis Federation (ITTF) approved Donic Stress table tennis net set (Donic, Voelklingen, Germany) was mounted onto the ITTF approved Delhi 25 table tennis table (Donic, Voelklingen, Germany). The standard D40+ 3 stars table tennis balls (Double Happiness, Shanghai, China) were loaded into the Newgy Robo-Pong 2050 (Newgy Industries, Inc., Hendersonville, TN, USA) to project balls to 2 oscillator positions (positions 15 and 5) for forehand and backhand topspin drives (Figure 2). Valid returns were considered when the balls landed diagonally on the lower half of the table near the robot (0.80 m × 0.76 m).

### 2.4. Experimental Protocol

All participants were given 2 min to warm up the forehand and backhand topspin drives by rallying 30 consecutive forehand topspin drives followed by another 30 consecutive backhand topspin drives. This sequence, which replicated a table tennis player warming up in competitions prior to the match, allowed participants to familiarize themselves with the height, speed, and spin of the balls projected by the robot when they were wearing the 3DSuit. The tests started after concluding the familiarization period. The order of executing forehand or backhand topspin drive first was randomly assigned for each participant. They performed a total of 30 trials (3 sets of 10 consecutive drives) for forehand topspin drives and 30 trials (3 sets of 10 consecutive drives) for backhand topspin drives. Only the trials with projected balls landed diagonally on the lower half of the table within the targeted zone (0.80 m × 0.76 m) were deemed valid (Figure 2b). All invalid trials were discarded. Since all participants had executed at least 10 valid trials in both forehand and backhand topspin drives, we standardized to analyze the first 10 valid trials per subject for good consistency.

### 2.5. Data Processing

#### 2.5.1. Filtering Raw Data

Joint angle data were recorded using the AnimaDemo program. The raw data were exported as .bvh files and then low-pass filtered at 8 Hz using the 4th order zero-lag Butterworth filter in MATLAB (MATrix LABoratory; R2017b, Mathworks, Natick, MA, USA) to smooth and remove random errors. Subsequently, each participant’s joint angle data were time normalized (0 to 100%) to obtain an average angle-time history of the 10 trials for the shoulder joint in the frontal plane, elbow joint in the sagittal plane, and wrist joint in the frontal plane (Figure 3).

#### 2.5.2. Extracting Key Variables

Each table tennis stroke can be split into different phases. Positions 1 to 2 refer to the preparation phase, positions 2 to 3 refer to the execution phase, and positions 3 to 4 refer to the follow-through phase (Figure 3). For each valid trial, the following variables were extracted:Minimum angle—the minimum angle for shoulder, elbow, and wrist joints is maximum shoulder abduction, elbow flexion, and wrist extension, respectively;Final angle—the angle at the end of the follow-through phase of each drive;Range of motion (ROM)—t he excursion from minimum to final angles of each drive.

The average values of 10 trials were calculated per participant in each type of drives. Group mean and standard deviation for the AB and para-players in different classifications were also determined.

## 3. Results

Descriptive data and graphical comparison of angle-time histories are presented to illustrate the different in movement patterns among AB and wheelchair players in varies classes (Figure 4). No inferential statistical analysis was performed owing to the small number of para-players in each sub-group. Data are expressed as mean (standard deviation).

### 3.1. Forehand Topspin Drives

AB demonstrated the greatest ROM for all joints of the forehand topspin drives (Table 2). All participants displayed similar forehand shoulder kinematics, showing abduction in the preparation phase, followed by adduction in the execution and follow-through phases (Figure 4). However, the forehand elbow and wrist joint motions differed substantially between AB and those with physical impairments. Noticeable differences in the upper limb kinematics among the different classes of wheelchair players were also observed. For instance, the wrist angles of C1 differed substantially from those of C2 (fixed at approximately 15°) and C3 (fixed at approximately 20°).

### 3.2. Backhand Topspin Drives

The backhand shoulder angles differed between AB and wheelchair players, while the profiles were similar among the different wheelchair classification groups (Figure 4). For the elbow and wrist joints, the angles varied to a large extent among C1 to C3 sub-groups and the AB players. Generally, wheelchair participants’ elbow joint motions followed the U-shape pattern while AB’s elbow joint motion followed the inverted U-shape pattern. Lastly, C1 exhibited a U-shaped wrist joint motion pattern, while AB showed an inverted U-shaped wrist joint motion pattern. In contrast, the wrist joint angle remained almost unchanged in C2 (approximately 10°) and C3 (approximately 0°) throughout the backhand drive.

## 4. Discussion

This study applied IMU sensors to profile the kinematics of the playing limb in table tennis players with and without physical disabilities. Shoulder, elbow, and wrist joint angles were compared among AB and wheelchair players in different classifications while performing the forehand and backhand topspin drives. To the best of the authors’ knowledge, the present study is the first to comprehensively report the upper limb kinematics of wheelchair table tennis players while executing table tennis strokes. The results showed clear differences between the AB and wheelchair players, especially in the elbow and wrist. Among the wheelchair table tennis players, noticeable variations in techniques were also observed in different disability classes.

Using portable IMU sensors, the AB’s shoulder and elbow joint angles observed in this study were similar to those reported by Xia and colleagues [35] who used a 10-camera motion capture system to examine the kinematics of elite table tennis players. Therefore, this study provided evidence that IMU sensors can capture table tennis techniques and movements like video or optical cameras with superior portability, affordability, and convenience (easy-to-use). The IMU sensors were a vital factor to increase para-players’ participation in this study because the sensors were brought to the para-players’ usual training venue so that trials can be conducted during their training. In addition, using the sensors would help to reduce barriers, such as inaccessibility, not wheelchair friendly facilities, etc., for the para-players when they are required to commute to remote laboratories with bulky motion capture system. We acknowledge that there are also disadvantages of the 3DSuit. For example, participants’ body shapes and sizes vary and therefore the 3DSuit may not fit well in all individuals. In addition, IMU sensors are sensitive to magnetic disturbances. Since the wheelchair, table tennis table, and net post contain metallic materials that can distort and interfere with measurement, additional step of magnetic deviation compensation was taken to correct for magnetic distortions. In comparison, camera-based motion capture systems offer greater flexibility in accommodating different body sizes and are unaffected by magnetic interference.

As with other wearable devices and compression garments, there will be some degrees of movement impediment resulting from wearing the 3DSuit. While we cannot quantify how the 3DSuit may have affected natural table tennis movements, we expect a small effect because the compression shirt and leggings are generally comfortable and flexible. In addition, the sensors were light and hence would not add much weight to the players’ limbs. Thus, we believe that the table tennis kinematic data measured using the gyroscopic 3DSuit inertial motion capture system are a good representation of the players’ usual techniques.

Findings from the present study provide empirical evidence to confirm the previous speculation that para-table tennis players would use alternative motion patterns to compensate for their physical impairments [32]. This is similar to other sports whereby para-athletes using alternative motion patterns to compensate for their physical disabilities [36,37]. For example, Bjerkefors and colleagues [36] found that elite para- and AB kayakers had distinct kayak paddling techniques by capitalizing on the upper extremities (e.g., greater maximum shoulder abduction, extension, and flexion and rotation ROM). In the present study, the shoulder kinematics in wheelchair and AB players were similar when executing the forehand drives. The elbow and the wrist movement patterns were distinct among AB and various Classes of wheelchair sub-groups. Unlike AB athletes who performed the table tennis drives with all three joints, C2 and C3 athletes exhibited very limited wrist motion in both forehand and backhand drives. This is likely due to wheelchair participants’ weak grip and wrist flexors [38], and an elastic bandage was wrapped around their hands to secure the playing hand’s racket firmly. The weak hand grip and the tight bandaging are plausible reasons explaining the small ROM in the wrist joint angle in C2 and C3 athletes. One of the three Class 1 player had full function of the wrist joint because he suffered from amputation where he only had his playing limb with two fingers and a prosthetic thumb (i.e., both lower extremities with above-the-knee amputation and left transhumeral amputation). The remaining two Class 1 players suffered from spinal cord injury due to accident where they have almost no function of the wrist joint. Such functional limitations can explain why the wrist joint angles for Class 1 players differed from AB and Classes 2 and 3 players. In addition, the para participants’ joint motion patterns resemble a “scooping” action where the arm, forearm, and hand rotate about the shoulder joint as an almost rigid segment due to the loss of elbow and wrist links in the kinetic chain. This contrasts with AB, who follow the proximal-to-distal sequencing of the playing limb [39,40].

No previous studies have examined wheelchair table tennis biomechanics while executing dynamic sport-specific strokes. This present study reports novel results that the playing limb kinematics differed not only between AB and para-athletes but also among different wheelchair disability classes. Since, our wheelchair participants were elite athletes competing at the national and international levels, it is believed that they had adapted optimal movement strategies to generate speed and power while maintaining balance to compensate for their respective impairments. Making direct references or replicating AB athletes’ movement patterns can lead to detrimental outcomes such as poor performance and injury. Given the movement strategies of para-athletes likely to be different from their AB counterparts, they should adapt and modify the movement patterns to suit their nature of physical impairment and sporting requirements for successful performance. The differences between AB and para-athletes have also been acknowledged in other sports. For instance, para-swimmers favored the parabolic fast start strategy in 400-m freestyle swimming, while the AB swimmers favored only the fast start strategy [41]. Thus, it could be inappropriate to assume that para-table tennis players should adapt AB table tennis skills and techniques completely.

Considering the marked differences in playing techniques observed in the present study, coaches should not ask para-athletes to simply replicate the skills and techniques used by AB table tennis players. Countless movement strategies exist, and some are more favorable depending on the interacting constraints of an individual’s abilities, task requirements, and environmental conditions. The ideal technique for para and AB players will very likely be distinctive even under the same condition such as returning a ball at a certain speed towards a certain direction. Traditional table tennis training for the AB players has greatly emphasized footwork training on how to move around the table efficiently [42]. However, this type of training is not feasible for para-players, especially for Classes 1–3 players competing in a wheelchair. As such, ball placement training is more relevant for Classes 1–3 players because the functional reach of the para-players is a crucial characteristic in para gameplay [23]. For example, the greater the functional reach of a para-player, the higher the probability covering a larger area of the table tennis table to return more rallies to their opponent. Based on the findings of this study, coaches of para-table tennis players should customize their training plan tailored to improve para-players’ ball placement during gameplay. For instance, coaches should teach para-players how to target the balls to land on their opponent table, especially at the front of the net and at the extreme sides (left and right) of the table by using their hand and forearm (i.e., wrist and elbow joints) to control their racket angle and amount of strength needed. Future research warrants the investigation for desirable training methods and optimal techniques for para-table tennis players from different classes, with various types and degrees of impairment.

There are a few limitations to the current study. Firstly, at a starting point on wheelchair table tennis, kinematic data were only obtained from the upper extremities of the playing limb. Future studies could investigate the trunk movement of the para-players as well as the manipulation of the wheelchair. Secondly, the sample size of each disability class was rather small due to the limited number of para-players available in Singapore. In the future, studies are recommended to include more para-players as well as those from the standing classes (i.e., Classes 6–10) who may adopt different techniques compared to wheelchair players. Finally, all para-table tennis players recruited in our study had a disability due to an accident later in adulthood. They picked up table tennis much later than able-bodied players who started training at a young age. The disparity in the number of years of experience between able-bodied and para-players may have affected the maturity of their table tennis technique.

## 5. Conclusions

The study revealed that wheelchair table tennis players exhibited distinctly different upper limb kinematics compared with AB participants while executing the forehand and backhand topspin drives. For the wheelchair table tennis players, noticeable variations in techniques were also observed among different disability classes. The differences were more pronounced in the elbow and wrist joints than the shoulder joint. The marked differences in techniques could be due to functional limitations associated with the wheelchair players’ physical impairment. Thus, coaches should be aware that traditional table tennis training programs targeting skills and techniques for AB players are unsuitable for para-players. Communication among para-players, coaches, and medical professionals are critical to better understand the players’ physical abilities and functional limitations. Future research can further investigate how to customize training programs and to optimize para-players’ movement strategies for successful performance.

## Figures and Tables

**Figure 1 sensors-21-08303-f001:**
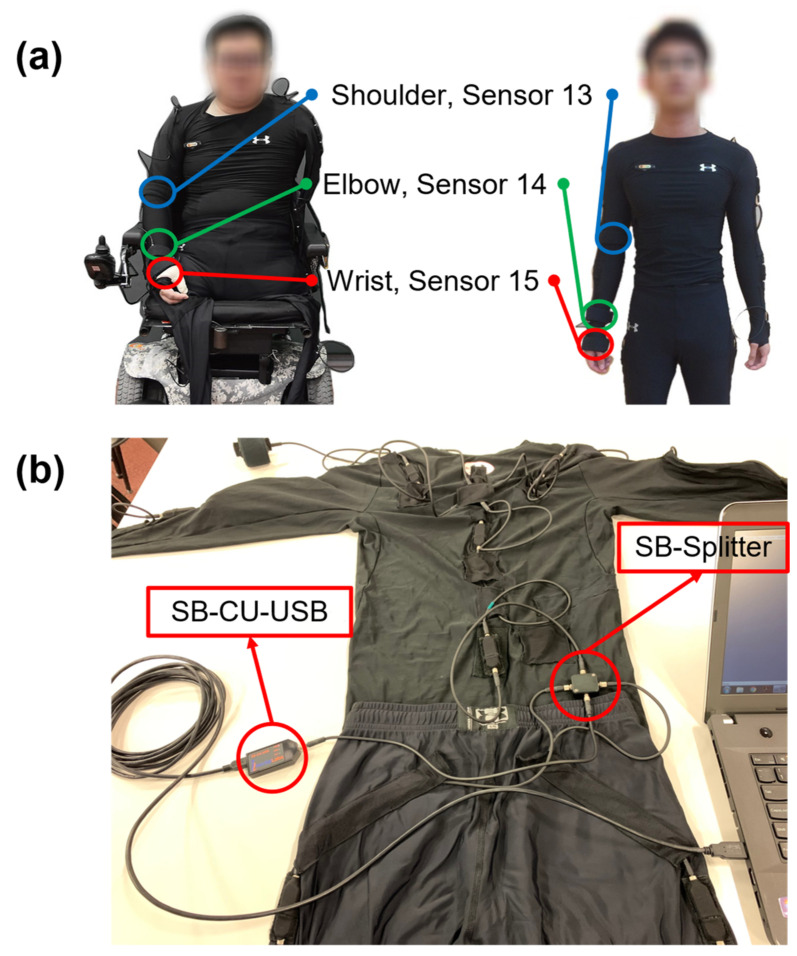
(**a**) The 3DSuit inertial motion capture system with the shoulder, elbow, and wrist sensors on wheelchair (**left**) and able-bodied (**right**; AB) table tennis players; (**b**) SB-Splitter and SB-CU-USB configuration for connecting the long sleeve compression shirt and leggings.

**Figure 2 sensors-21-08303-f002:**
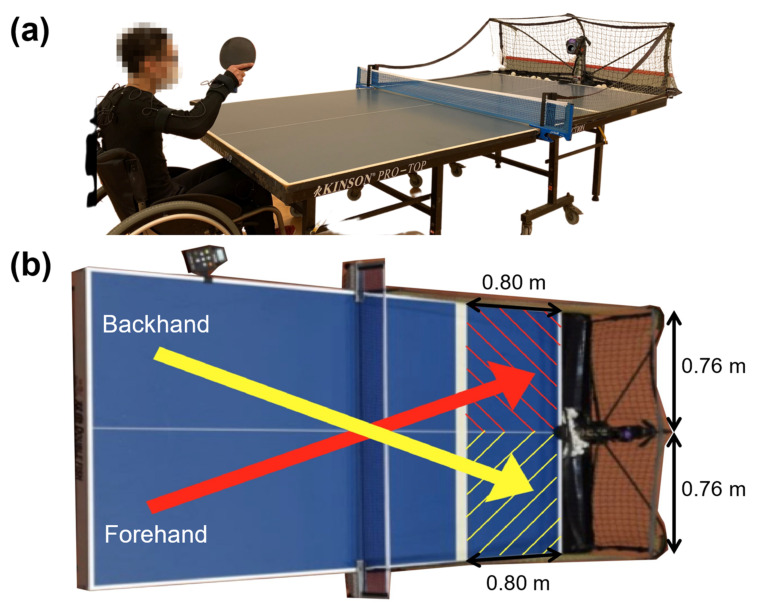
(**a**) Experimental setup using the table tennis table, net set, balls, and Newgy Robo-Pong 2050; (**b**) dimensions (0.80 m × 0.76 m) for valid forehand (red) and backhand (yellow) topspin drives. Balls are projected to oscillator positions 15 and 5 for forehand and backhand topspin drives, respectively.

**Figure 3 sensors-21-08303-f003:**
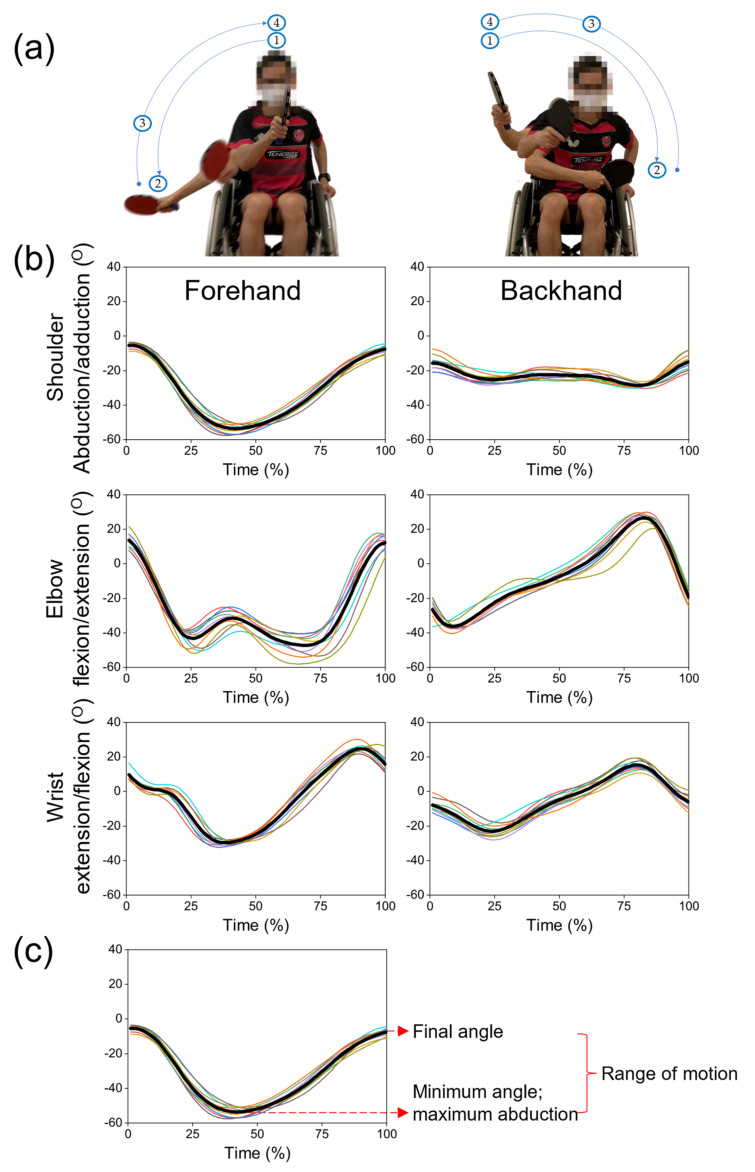
(**a**) All participants performed the forehand (**left**) and backhand (**right**) topspin drives where positions 1 to 2 refer to the preparation phase, positions 2 to 3 refer to the execution phase, and position 3 to 4 refer to the follow-through phase; (**b**) time-normalized joint angles for forehand and backhand topspin drive where each figure presents 10 analyzed trials for one participant with the solid black line representing the mean value; (**c**) extracting key variables of interest for each drive where minimum angle, final angle, and range of motion (ROM) for each joint are determined.

**Figure 4 sensors-21-08303-f004:**
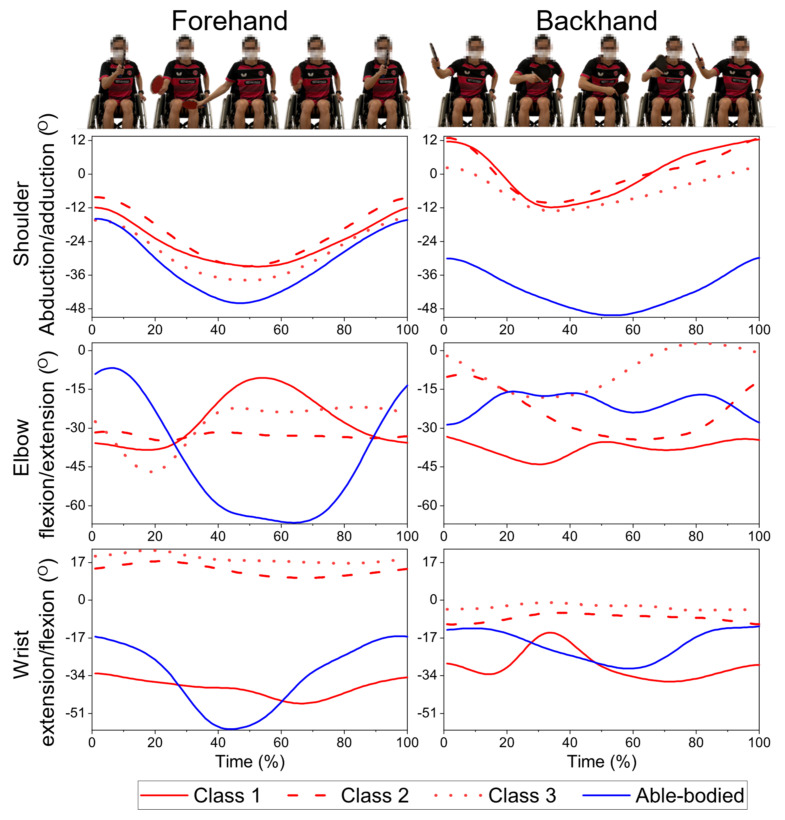
Shoulder, elbow, and wrist joint angle-time histories among classifications 1 (*n* = 3), 2 (*n* = 3), 3 (*n* = 4), and able-bodied (*n* = 9) participants while performing the forehand topspin drives (**left**) and backhand topspin drives (**right**). A positive and increasing angle indicates shoulder adduction, elbow extension, and wrist flexion, while a negative and decreasing angle indicates shoulder abduction, elbow flexion, and wrist extension.

**Table 1 sensors-21-08303-t001:** Characteristics of 19 male wheelchair (classes 1 to 3) and able-bodied table tennis players.

Classification		Class 1	Class 2	Class 3	Able-Bodied
*n*		3	3	4	9
Age (years)		52.3 (9.1)	44.0 (5.6)	50.3 (17.2)	23.1 (1.6)
Experience (years)		3.3 (0.6)	4.7 (1.5)	3.0 (1.4)	13.4 (2.6)
Playing hand		Right-handed	2 Right-handed 1 Left-handed	Right-handed	Right-handed
Grip style		Shakehand	Shakehand	Shakehand	Shakehand
Racket Type	Front	2 Inverted 1 Long pimples	2 Inverted 1 Long pimples	2 Inverted 2 Mid pimples	Inverted
	Back	1 Inverted 2 Long pimples	2 Inverted 1 Long pimples	1 Inverted 1 Short pimples 2 Long pimples	7 Inverted 1 Short pimples 1 Long pimples

Classes 1 to 3 are wheelchair participants, *n* = 10. Front refers to the front side (forehand) of the racket and Back refers to the back side (backhand) of the racket. Data are presented as mean (SD).

**Table 2 sensors-21-08303-t002:** Shoulder, elbow, and wrist joint angles between Classes 1, 2, 3, and AB participants while performing table tennis forehand and backhand topspin drives.

Joint	Minimum Angle (°)				Final Angle (°)				ROM (°)			
	C1	C2	C3	AB	C1	C2	C3	AB	C1	C2	C3	AB
**Forehand**												
Shoulder	32.9 (11.9)	32.9 (8.7)	38.4 (5.4)	48.1 (8.5)	11.8 (18.7)	8.2 (5.8)	15.3 (9.1)	16.1 (12.7)	21.1 (6.8)	24.7 (4.9)	23.0 (11.2)	32.0 (15.1)
Elbow	39.2 (9.5)	37.9 (10.8)	49.9 (21.7)	70.3 (24.5)	35.7 (8.4)	33.0 (14.9)	25.3 (43.5)	12.2 (26.0)	3.5 (1.2)	5.0 (4.1)	24.7 (31.0)	58.0 (31.8)
Wrist	50.7 (11.1)	8.4 (19.5)	16.0 (25.1)	69.1 (49.0)	34.6 (7.8)	14.2 (16.6)	18.9 (22.6)	16.6 (17.6)	16.1 (3.3)	5.8 (4.3)	2.9 (2.6)	52.5 (37.9)
**Backhand**												
Shoulder	17.0 (31.2)	10.9 (1.1)	13.7 (13.5)	54.4 (13.8)	12.5 (10.7)	13.0 (14.6)	2.7 (17.1)	29.5 (10.9)	29.6 (39.3)	23.9 (15.0)	16.4 (9.4)	24.9 (15.8)
Elbow	46.5 (16.0)	34.6 (44.3)	19.3 (18.8)	40.8 (11.4)	34.7 (9.8)	11.3 (18.8)	1.3 (27.8)	28.2 (6.4)	11.8 (6.2)	23.2 (25.6)	18.0 (11.4)	12.6 (14.1)
Wrist	39.4 (2.1)	12.8 (13.4)	6.7 (29.1)	45.1 (49.7)	29.1 (5.5)	10.9 (12.5)	4.2 (29.1)	11.7 (20.0)	10.3 (3.3)	1.9 (2.0)	2.5 (1.4)	33.3 (50.4)

Data are expressed as mean (standard deviation). ROM denotes range of motion. Shoulder, elbow, and wrist minimum joint angles refer to maximum shoulder abduction, elbow flexion, and wrist extension, respectively. Joint ROMs are excursions from maximum shoulder abduction, elbow flexion, and wrist extension to the end of the drives.

## Data Availability

Data are available at the NIE Data Repository https://researchdata.nie.edu.sg/privateurl.xhtml?token=ef175f76-1493-448f-873e-482101b95573.

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
