# Peer review of "Measuring Upper Limb Kinematics of Forehand and Backhand Topspin Drives with IMU Sensors in Wheelchair and Able-Bodied Table Tennis Players"

_sensors, 2021, doi:10.3390/s21248303_

Round 1

Reviewer 1 Report

I usually have a lot of comments on the manuscripts, but this paper was really interesting and not complicated. Oh, maybe two small remarks:

1) authors could add short explanation how the angles were maesured by sensors 13, 14 and 15.

2) What does it mean that "the system had a high accuracy of 1% for position" (line122)?

Author Response

I usually have a lot of comments on the manuscripts, but this paper was really interesting and not complicated. Oh, maybe two small remarks:

  • Thank you for the positive feedback. Your time and effort to help us improve our work are much appreciated.

1) Authors could add short explanation how the angles were measured by sensors 13, 14 and 15.

  • Thank you for the suggestion. We have added a more detailed explanation in the manuscript:

“Before commencing all trials, participants were required to face south in an initialisation posture to calibrate the system. This prerequisite posture entailed them standing in the anatomical position with face directed forward, arms at the side with palms facing forward, and feet shoulder-width apart to reset sensors 13-15 to 0°. Subsequently, participants moved to their ready position before executing the table tennis techniques.” (Lines 135 to 139)

2) What does it mean that “the system had a high accuracy of 1% for position” (line122)?

Sorry for the oversight. We meant to refer to angle measurement only and not position. We have since revised the sentence: “According to the manufacturer’s specifications, the system had a high accuracy of 1° for angle measurement.” (Lines 121-122).

Reviewer 2 Report

The authors compared 12 kinematics of the shoulder, elbow and wrist joints between fit players (AB) and in wheelchairs from 13 different classifications. This study aims at a better understanding of the biomechanics of table tennis players. For this, nineteen participants were taken into account (AB, n = 9; classification 1 (C1), n ​​= 3; C2, n = 3; C3, n 14 = 4) for which 10 drives were executed. topspin with forehand and backhand. Shoulder abduction / adduction, elbow flexion / extension 15, wrist extension / flexion, respective range of motion (ROM) and joint models 16 were obtained using inertial unit (IMU) sensors. The results showed clear differences in upper limb kinematics between fit and wheelchair players, especially at the elbow and wrist. For parachutists, there were also notable variations in techniques 19 between different classes of disabilities.

The introduction makes a good presentation of the current state of research in the field, so that in the end the authors can concisely present the purpose of the paper. The experimental part presents data on materials and methods, with a clear emphasis on the working protocol and the processing of experimental data. The results and discussions are well elaborated, involving rigorously represented figures and data tables. The conclusions are well drawn. The references are current and representative. The paper can be published in its current form.  

Author Response

  • Thank you for your positive feedback.

Reviewer 3 Report

Table1: Isn't the disparity of experience, in number of years, in the participants an obstacle to the study?
What is the impact of being right or left handed on the gesture of the pongist? Doesn't this interfere with your conclusions?

L116:How to make sure of the good positioning of the sensors as they are not on the skin but on a garment?

Figure 1: Images too dark, sensors are not very visible
To what extent do the suits not impede the movement of the pongers?

L168:With all these tests, isn't there a risk of fatigue for the participants?

Figure 3: Figures b and c are blurred and not very legible.

L206:The explanation on how to determine the wrist positioning angles (relative to the median plane?) is not clear, an additional diagram would be helpful.

Table 2:Do the numbers in parentheses correspond to the SD? If so, why are they, in 16 different places, larger than the angle values? Explanations? Consequences on the measurements? Validity?

L258:IMU sensors, according to you, have only advantages ?
Isn't having a combination size for each patient a disadvantage compared to motion picture cameras?

The analysis of the results may not be thorough enough, one would expect to be able to draw more conclusions from this work. What kind of specific recommendations or training programs can be drawn from this work, specifically for table tennis? This aspect could be more detailed. A common training recommendation for all classes seems difficult to make given the difference in pathology between table tennis players. Therefore, the final interest of the study is somewhat diminished since the conclusions are not complete. Complete your conclusion.

Check the formatting of the bibliography, it is not homogeneous

Author Response

1) Table 1: Isn’t the disparity of experience, in the number of years, in the participants an obstacle to the study?

  • Thank you for your insights. All para-table tennis players recruited in our study had a disability due to an accident later in adulthood. Therefore, para-players picked up table tennis much later than able-bodied players who started training at a young age. We acknowledged that the disparity in the number of years of experience between able-bodied and para-players may have affected the maturity of their table tennis technique which is a limitation of the study. We have now discussed this issue under the limitation section:

“Finally, all para-table tennis players recruited in our study had a disability due to an accident later in adulthood. They picked up table tennis much later than able-bodied players who started training at a young age. The disparity in the number of years of experience between able-bodied and para-players may have affected the maturity of their table tennis technique. (Lines 356 to 360)

2) What is the impact of being right or left-handed on the gesture of the pongist? Doesn’t this interfere with your conclusions?

  • Thank you for pointing this out. In an actual game, there will be an impact when a right-handed player is competing against a left-handed player as the right-handed player will be using the forehand to rally with the backhand of the left-handed player. For our study, there is minimal impact as the ball was projected from a robot. Both right-handed and left-handed players are similar with forehand and backhand topspin drives We have also checked that the kinematic profiles of the left-handers are comparable with the right-handers.

Figure 1. Forehand shoulder abduction/adduction for all para-players (n = 10). RH refers to right-handed players, and LH refers to left-handed players.

3) L116: How to make sure of the good positioning of the sensors as they are not on the skin but on a garment?

  • Thank you for pointing this out. To ensure that the sensors were worn on all participants correctly and consistently, one researcher from our team followed the standard procedures to insert the three sensors (shoulder, elbow, and wrist) into the movable pocket and checked if the actual sensor positions fitted the appropriate anatomical landmarks. After that, the movable pockets were secured with Velcro straps on top of the 3DSuit (i.e., Under Armor long sleeve compression tights). We have added this sentence in the revised manuscript:

“After checking the position of each sensor, which should be on the appropriate anatomical landmark, a Velcro strap was tightened to secure the sensors.” (Lines 126-128).

4) Figure 1: Images too dark, sensors are not very visible

  • We agreed that it is hard to see because both the suit and the sensors are in black. To help readers locate the locations of the three sensors, we used colour circles (blue for shoulder, green for elbow, red for wrist) with annotations in Figure 1. We hope that readers will have a good overview of how the IMU suit looks like when worn on a participant, and the approximate locations of the sensors.

To what extent do the suits not impede the movement of the pongers?

  • As with other wearable sensors and compression garments, there will be some degrees of movement impediment. While we cannot quantify the extent to how the 3DSuit may have affected natural table tennis movements, we expect a small effect. This is because the 3DSuit uses a Men’s Under Armor HeatGear® long sleeve compression shirt and leggings, which are generally comfortable and flexible. In addition, the sensors were light, and hence that did not add much weight to the players’ limbs. We have now expanded our Discussion to include the possible movement impediment:

“As with other wearable devices and compression garments, there will be some degrees of movement impediment resulting from wearing the 3DSuit. While we cannot quantify how the 3DSuit may have affected natural table tennis movements, we expect a small effect because the compression shirt and leggings are generally comfortable and flexible. In addition, the sensors were light and hence would not add much weight to the players’ limbs. Thus, we believe that the table tennis kinematic data measured using the gyroscopic 3DSuit inertial motion capture system are a good representation of the players’ usual techniques.” (Lines 277 to 284)

5) L168: With all these tests, isn’t there a risk of fatigue for the participants?

  • Thank you for pointing this out. During the experiment sessions, the overall duration of all trials for each participant was only about 5 to 10 minutes, which was not considered a long period compared with their training routines. Furthermore, all participants were committed to training regularly for 3 hours per session and at least two sessions per week by the time of the experiment. Hence, the test results should not be affected by fatigue.

6) Figure 3: Figures b and c are blurred and not very legible.

  • Thank you for your suggestion. We have improved the resolution and font size of all figures, including Figure 3.

7) L206: The explanation on how to determine the wrist positioning angles (relative to the median plane?) is not clear, an additional diagram would be helpful.

  • Thank you for pointing this out. The sensors were set at zero degrees in the standardised calibration posture facing south. Subsequently, wrist angles in the frontal plane were extracted for analysis. We have expanded our Methods description in the revised manuscript:

“Before commencing all trials, participants were required to face south in an initialisation posture to calibrate the system. This prerequisite posture entailed them standing in the anatomical position with face directed forward, arms at the side with palms facing forward, and feet shoulder-width apart to reset sensors 13-15 to 0°.” (Lines 135-138)

“Subsequently, each participant’s joint angle data were time normalised (0 to 100%) to obtain an average angle-time history of the 10 trials for the shoulder joint in the frontal plane, elbow joint in the sagittal plane, and wrist joint in the frontal plane (Figure 3).”  (Lines 184-187).

8) Table 2: Do the numbers in parentheses correspond to the SD? If so, why are they, in 16 different places, larger than the angle values? Explanations? Consequences on the measurements? Validity?

  • Yes, our data are expressed as mean (standard deviation). We have checked our data and confirmed that the results are correct. SD is larger than the mean angle due to the high inter-individual differences in the group, especially for the para-athletes with varying degrees of functional disabilities. For instance, Figure 4 illustrates that the wrist angle can range from +20 degrees to -40 degrees. The large SD is not a validity issue of the IMU unit but reflects the diverse techniques employed by the table tennis players.

9) L258: IMU sensors, according to you, have only advantages? Isn’t having a combination size for each patient a disadvantage compared to motion picture cameras?

  • We agree with the reviewer that there are advantages and disadvantages of each type of sensors. We have since revised the manuscript to provide a more balanced view. The changes are:

“We acknowledge that there are also disadvantages of the 3DSuit. For example, participants’ body shapes and sizes vary and therefore the 3DSuit may not fit well in all individuals. In addition, IMU sensors are sensitive to magnetic disturbances. Since the wheelchair, table tennis table, and net post contain metallic materials that can distort and interfere with measurement, additional step of magnetic deviation compensation was taken to correct for magnetic distortions. In comparison, camera-based motion capture systems offer greater flexibility in accommodating different body sizes and are unaffected by magnetic interference.” (Lines 269-276)

10) The analysis of the results may not be thorough enough, one would expect to be able to draw more conclusions from this work. What kind of specific recommendations or training programs can be drawn from this work, specifically for table tennis? This aspect could be more detailed. A common training recommendation for all classes seems difficult to make given the difference in pathology between table tennis players. Therefore, the final interest of the study is somewhat diminished since the conclusions are not complete. Complete your conclusion.

  • Thank you for the insightful comments and suggestions. We have added more practical recommendations for training in the Discussion:

“Traditional table tennis training for the AB players has greatly emphasised footwork training on how to move around the table efficiently [42]. However, this type of training is not feasible for para-players, especially for Classes 1-3 players competing in a wheelchair. As such, ball placement training is more relevant for Classes 1-3 players because the functional reach of the para-players is a crucial characteristic in para gameplay [23]. For example, the greater the functional reach of a para-player, the higher the probability covering a larger area of the table tennis table to return more rallies to their opponent. Based on the findings of this study, coaches of para-table tennis players should customise their training plan tailored to improve para-players’ ball placement during gameplay. For instance, coaches should teach para-players how to target the balls to land on their opponent table, especially at the front of the net and at the extreme sides (left and right) of the table by using their hand and forearm (i.e., wrist and elbow joints) to control their racket angle and amount of strength needed.” (Lines 333 to 346)

11) Check the formatting of the bibliography, it is not homogeneous

  • Thank you for pointing this out. We have checked and revised the reference list to ensure it is consistent with the journal’s requirements.
